# Learning Robotic Manipulation Policies from Point Clouds with Conditional Flow Matching

**Eugenio Chisari, Nick Heppert, Max Argus, Tim Welschehold,**
**Thomas Brox, Abhinav Valada**
Department of Computer Science, University of Freiburg, Germany
http://pointflowmatch.cs.uni-freiburg.de

**Abstract:** Learning from expert demonstrations is a promising approach for training robotic manipulation policies from limited data. However, imitation learning algorithms require a number of design choices ranging from the input modality, training objective, and 6-DoF end-effector pose representation. Diffusion-based methods have gained popularity as they enable predicting long-horizon trajectories and handle multimodal action distributions. Recently, Conditional Flow Matching (CFM) (or Rectified Flow) has been proposed as a more flexible generalization of diffusion models. In this paper, we investigate the application of CFM in the context of robotic policy learning and specifically study the interplay with the other design choices required to build an imitation learning algorithm. We show that CFM gives the best performance when combined with point cloud input observations. Additionally, we study the feasibility of a CFM formulation on the $SO(3)$ manifold and evaluate its suitability with a simplified example. We perform extensive experiments on RLBench which demonstrate that our proposed PointFlowMatch approach achieves a state-of-the-art average success rate of 67.8% over eight tasks, double the performance of the next best method.

**Keywords:** Imitation Learning, Manipulation, Conditional Flow Matching

## 1 Introduction

Imitation learning (IL) is the widely studied problem of training policies from a given set of expert demonstrations [1, 2, 3]. In recent years, imitation learning has gained popularity in the robot learning community, as leveraging the prior knowledge of the expert demonstrator allows training complex behaviors with small amounts of data. The primary approach to learning an IL policy is Behavior Cloning (BC) [4, 5], where a deterministic mapping from state to actions is learned in a supervised manner from the available data. While BC has achieved significant success for different tasks, robot policy learning remains a challenging problem, given the requirement of high precision, the sequential correlation (i.e. not i.i.d.) of data, and the multimodality of the action distribution, which all add complexity compared to other supervised learning problems.

Recently, generative models have been demonstrated to be effective at tackling some of these challenges. Most prominently, Diffusion Policy [6] adopts a score-matching formulation of generative diffusion models. The forward diffusion process starts with expert robot trajectories and gradually adds Gaussian noise until the signal approximates pure noise. The denoising process reverts these steps and it is used as a training signal for the model. This is a stochastic process that results in Gaussian conditional probability paths mapping Gaussian noise to data, with specific choices of mean and standard deviation [7, 8]. The authors show that diffusion policies can handle multimodal action distributions and to directly predict long sequences of actions.

Diffusion policies sparked strong interest from the community and led to many exciting results in robotic manipulation applications [9, 10]. Nevertheless, relying on diffusion models also has some

8th Conference on Robot Learning (CoRL 2024), Munich, Germany.

disadvantages: diffusion models need to explicitly define an iterative forward diffusion process, which inherently defines the final noisy distribution and the probability path the model learns to denoise along. In the typical case of Gaussians, a closed-form solution is available, enabling us to directly generate fully noised and intermediate, partially noised, samples. In turn, in the cases where no closed-form solution for the forward diffusion process is available, training time will increase [11]. To overcome these limitations, Conditional Flow Matching (CFM) has been proposed as an efficient generalization of diffusion models [12, 13, 11]. CFM is a simulation-free approach, i.e. it starts directly from noise without requiring a forward diffusion process. It is, therefore, more general, since it allows transporting any starting probability distribution into the data distribution.

Inspired by recent flow-based generative models, we propose PointFlowMatch, a novel imitation learning algorithm for robotic manipulation. PointFlowMatch uses point cloud observations that prove to be more effective than images [9, 10] and builds upon a CFM formulation to learn the distribution of the expert robot trajectories. As CFM is able to model arbitrary probability paths, it also allows formulating the regression on the $\mathbb{R}^3 \times SO(3)$ manifold. We evaluate the performance of our proposed method on the popular RLBench benchmark [14] and compare it against strong recent baselines with both image and point cloud observations: Diffusion Policy [6], 3D Diffusion Policy [9], ChainedDiffuser [10], and AdaFlow [15]. In summary, this paper makes the following main contributions:

- PointFlowMatch, a novel method based on the recent conditional flow matching framework to train robotic imitation learning policies from point clouds.
- An investigation of two different approaches to handle 3D rotations in the context of CFM for policy learning.
- Extensive evaluations against recent state-of-the-art baselines and an ablation study of our main design choices.
- We make the code, models, and videos publicly available at `http://pointflowmatch.cs.uni-freiburg.de`.

## 2 Related Work

Generative models for imitation learning in robotic manipulation have attracted significant interest. In the following, we highlight recent advances in this topic.

**Generative Models for Policy Learning**: One interesting class of generative models explored for robotic policies is the Action Chunking Transformer (ACT) [16]. It adopts a CVAE [17] structure, where the encoder predicts a latent variable $z$ and the decoder outputs the action sequence. A different approach is taken by Florence et al. [18], who propose Implicit Behavior Cloning (IBC). IBC defines the distribution over action as an Energy-Based Model (EBM) [19]. This approach inherently handles multimodal distributions but is unstable to train due to the need for negative sampling when computing the loss. An early attempt to apply diffusion models to robotic applications is presented in Diffuser [20], where the authors investigate the use of diffusion models from the perspective of planning. Refining this idea, Chi et al. [6] propose the seminal work Diffusion Policy (DP). DP inherits the advantages of IBC (handling multimodal action distributions) while avoiding its downsides. Instead of learning an energy function, DP learns the gradient of the action distribution, which exhibits better training stability.

**Diffusion Policy from Point Clouds**: The success of DP inspired numerous follow-up works. A key characteristic common to many of them is the choice of point cloud observation representations, instead of RGB images, as used in the original DP method. ACT3D [21] is a transformer model that predicts the next best waypoint, instead of a full trajectory. 3D-DP [9] learns to predict full trajectories, and it also adopts a modified MLP as an efficient point cloud backbone. ChainedDiffuser [10] proposes a hierarchical model where high-level key points are predicted via ACT3D and low-level trajectories connecting these points are generated via the standard trajectory diffusion. Another hierarchical model is HDP [22], which builds on PerAct [23] as a waypoint predictor and combines both joint and end-effector pose predictions. While hierarchical models achieve strong results, we

specifically investigate the performance of the underlying trajectory diffusion models and therefore focus our analysis on non-hierarchical single-task policies.

**Conditional Flow Matching**: Conditional Flow Matching (CFM), also called Rectified Flow (RF), is a recent generative model training technique that connects noise to data via straight paths [12, 13, 11]. The more established diffusion models can be seen as a special case of the CFM framework, which is more general and conceptually simpler. Very recently, CFM was demonstrated to be superior to diffusion models in various applications, ranging from image synthesis [24] to protein backbone generation [25]. In the context of robotics, AdaFlow [15] is an image-based policy trained with CFM, which also includes a variance estimation network to predict the variance of the current state and adjust the number of integration steps dynamically in order to reduce inference time. Concurrently to our work, Braun et al. [26] investigate the application of CFM to Riemannian manifolds, but do not provide a direct comparison with the standard Euclidean approach, while Rouxel et al. [27] show the application of CFM for shared autonomy teleoperation of a humanoid robot.

## 3 Technical Approach

We consider an imitation learning problem: given a dataset of $n$ expert demonstrations $D = \left\{ \left( s^{(i)}, a^{(i)} \right) \right\}_{i=1}^{n}$, the goal is to train a policy $\pi : \mathcal{S} \to \mathcal{A}$ that maps the observed states $s$ to the optimal actions $a$. Given the widespread accessibility of depth cameras, we assume that both RGB and depth observations are available to the policy. In the following sections, we detail our choices of observation and action spaces, training formulation, and model architecture.

### 3.1 Observation and Action Spaces

It has been demonstrated that, for robot policy learning, point cloud observations yield better results compared to RGB images [9]. These findings are supported by the fact that most recent state-of-the-art approaches [21, 22, 10] building upon DP adopt point clouds as their visual observation representations. Point cloud observations are effective as they directly encode the three-dimensional structure of a scene, and clearly separate geometric and semantic features, which are instead blended together in raw RGB images. This separation is especially helpful in the low data regimes common in robot policy learning. Therefore, in our approach PointFlowMatch we also adopt point clouds as the visual observation representation.

The input to our policy consists of the $T_{\text{obs}}$ last observations, which include both the robot state and processed point cloud, where $T_{\text{obs}}$ is a tuning hyperparameter. Point clouds are processed in the following manner. For each available camera, we use the intrinsics and extrinsics to project the observed depth values in a common 3D space, and we merge all observations into a single point cloud. Last, the final point cloud is computed by applying voxel-downsampling to attain uniform density, and by cropping all points outside the relevant workspace around the robot.

The action space of our model is 10-dimensional, consisting of end-effector position (3-dimensional), end-effector orientation (6-dimensional), and gripper open/close action (1-dimensional). We choose absolute position control following Chi et al. [6], which showed it to be more suited to action-sequence prediction compared to velocity control. We represent the gripper orientation as the 6D vector formed by truncating and flattening the relative rotation matrix, as proposed by Zhou et al. [28]. At test time, we employ a closed-loop receding horizon control strategy, i.e. we predict $T_{\text{pred}}$ horizon steps into the future, command only the first predicted step to the robot, and then repeat the prediction.

### 3.2 Conditional Flow Matching for Policy Learning

In CFM, for arbitrary samples $\boldsymbol{z}$, the prediction problem is formulated as an ordinary differential equation (ODE) of the form

$$\frac{\mathrm{d}}{\mathrm{d}t} \boldsymbol{z}(t) = \boldsymbol{v}(\boldsymbol{z}(t), t), \qquad \boldsymbol{z}(t = 0) \sim p_0, \tag{1}$$

where $p_0$ is the arbitrary start distribution, commonly chosen as random Gaussian noise with zero mean and unit variance, and $\boldsymbol{v} : \mathbb{R}^d \times [0, 1] \to \mathbb{R}^d$ is the velocity vector field that induces the

probability path from the start to the target distribution. The goal is to learn the appropriate vector field that transports the chosen probability distribution to the data distribution in unit time. The vector field $\boldsymbol{v}_\theta$ can be learned by supervision via the loss function

$$\min_{\boldsymbol{v}} \mathbb{E}_{\boldsymbol{z}_0 \sim p_0, \boldsymbol{z}_1 \sim p_1} \left[ \int_0^1 \|\boldsymbol{v}_{\text{gt}}(\boldsymbol{z}_1, \boldsymbol{z}_0) - \boldsymbol{v}_\theta(\boldsymbol{z}_t, t)\|_2^2 \ \mathrm{d}t \right], \tag{2}$$

where $v_{\text{gt}}$ is the choice of ground truth vector field we regress against. While this choice is not unique, CFM adopts the time-independent, straight line vector field $v_{\text{gt}} = \boldsymbol{z}_1 - \boldsymbol{z}_0$, which has the desirable property of producing straight probability paths [12, 11]. In practice, the vector field $\boldsymbol{v}_\theta$ is parametrized by a neural network and the loss in Eq. (2) can be minimized via mini-batch gradient descent. In the context of policy learning, the target samples $\boldsymbol{z}_1$ are the robot trajectories from the expert demonstration dataset. Once the model is trained, we can sample $\boldsymbol{z}_0$ from the start distribution $p_0$ and numerically integrate the ODE from Eq. (1) up to time $t = 1$ to generate new robot trajectories that follow our expert data distribution. An overview of the implementation of CFM for policy learning is shown in Algorithm 1.

---

**Algorithm 1** CFM for Policy Learning (Euclidean)

```
1  def train_step(batch):
2    obs, target_traj = batch
3    cond = obs_encoder(obs)
4    t = rand()  # uniform
5    z0 = randn()  # normal
6    z1 = target_traj
7    target_vel = z1 - z0
8    zt = z0 + t * target_vel
9    pred_vel = net(zt, t, cond)
10   loss = mse(pred_vel, target_vel)
11   return loss
12
13 def inference_step(obs):
14   cond = obs_encoder(obs)
15   z = randn()  # normal
16   for k in k_steps:
17     t = k / k_steps
18     dt = 1 / k_steps
19     pred_vel = net(z, t, cond)
20     z = z + pred_vel * dt
21   return z
```

**Algorithm 2** CFM for Policy Learning ($SO(3)$)

```
1  def train_step(batch):
2    obs, target_traj = batch
3    cond = obs_encoder(obs)
4    t = rand()  # uniform
5    z0 = randn_SO3()  # IGSO(3)
6    z1 = target_traj
7    target_vel = Log(Inv(z0) @ z1)
8    zt = z0 @ Exp(t * target_vel)
9    pred_vel = net(zt, t, cond)
10   loss = mse(pred_vel, target_vel)
11   return loss
12
13 def inference_step(obs):
14   cond = obs_encoder(obs)
15   z = randn_SO3()  # IGSO(3)
16   for k in k_steps:
17     t = k / k_steps
18     dt = 1 / k_steps
19     pred_vel = net(z, t, cond)
20     z = z @ Exp(pred_vel * dt)
21   return z
```

---

### 3.3 Conditional Flow Matching for Data in $SO(3)$

Our robot state representation includes the end-effector 3D orientation. Since 3D orientations live on the $SO(3)$ group's manifold, we need to pay particular attention to how we handle the orientation's prediction. There are two strategies we can employ to tackle this challenge.

*Euclidean Formulation*: The first strategy is to follow the standard CFM formulation in Euclidean space for both training and inference and project the resulting vector to the $SO(3)$ group's manifold only at the end of the inference process, to output a valid rotation matrix. This projection can be implemented as a Grahm–Schmidt process in the case of a 6D representation of rotations or as the SVD-based orthogonal Procrustes in the case of a 9D representation of rotations [29].

*$SO(3)$ Formulation*: The second strategy is to define the starting random distribution and target vector field directly on the 3D rotation manifold so that the entire resulting probability path lives on $SO(3)$. In particular, the random initial state is sampled as $\boldsymbol{z}(t=0) \sim \mathcal{IG}_{\text{SO}(3)}$, where $\mathcal{IG}_{\text{SO}(3)}$ is the isotropic Gaussian distribution on $SO(3)$. Since the 3D rotation group is a smooth manifold, its velocity at a specific point is expressed in the tangent space, denoted as $so(3)$. The mapping from the tangent vector space to the manifold is defined by the exponential map $\text{Exp}(\bullet)$, while the inverse mapping from the manifold to the tangent space is defined by the logarithm map $\text{Log}(\bullet)$. For

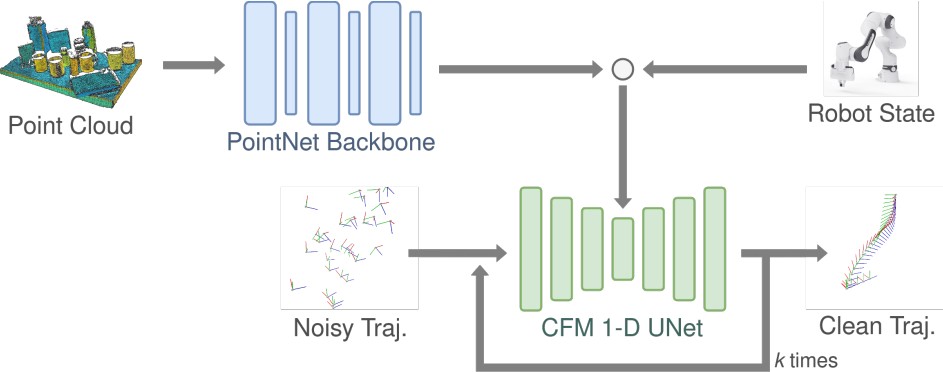

Figure 1: Diffusion and CFM are repeatedly applied to a noisy trajectory, thereby iteratively yielding a clean trajectory that can be executed on the robot. The generative models also take as input encoded observations.

a detailed overview of the topic, we refer the readers to [30]. For our CFM setting, the natural choice of target vector field is given by the tangent on the manifold that forms the geodesic interpolation between the start and target samples

$$z_t = z_0 \, \mathrm{Exp}\left(t \, v_{\mathrm{gt}}(z_1, z_0)\right), \quad \text{where} \quad v_{\mathrm{gt}}(z_1, z_0) = \mathrm{Log}\left(z_0^{-1} z_1\right). \tag{3}$$

At test time, we can integrate the predicted velocity vectors in tangent space along the manifold geodesic to solve the ODE in Eq. (1) and generate end-effector orientations that directly lie in $SO(3)$. The implementation of the $SO(3)$ formulation of CFM for policy learning is shown in Algorithm 2. While the latter approach has been studied in recent works such as Riemannian Flow Matching [31] and FoldFlow [25], a comparison of the two strategies in the context of policy learning is still lacking. In our evaluation (Appendix A.1), we investigate both strategies and provide the counter-intuitive insights that the first, simpler approach might be preferable for policy learning in robotics.

### 3.4 Model Architecture and Training Setup

To encode the point cloud observation into low-dimensional features, we use a modified version of PointNet [32]. Compared to the original architecture, we remove both T-nets used for input and feature transformation, see [33]. These are used to improve rotation and translation invariance, which is desirable for classification and segmentation but not for robotics tasks. The model used to de-noise trajectories is a conditional 1D U-Net, similar to [6]. This model takes as input the random noise samples and the condition information and returns the sample velocity in the case of CFM and the noise $\epsilon$ in the case of diffusion. For an overview, see Fig. 1. The condition information consists of the concatenation of the robot's proprioceptive state (i.e. end-effector poses) as well as the encoded visual observation, and it is added to the bottleneck state of the U-Net model.

We train our model with the AdamW optimizer and a learning rate of $3e^{-5}$ and weight decay of $1e^{-6}$. We apply cosine annealing of the learning rate and linear warmup of 5000 steps. We use a batch size of 128 and apply EMA on the weights. The input point clouds are downsampled to 4096 points and the target robot trajectories are subsampled with a factor of 3.

## 4 Experimental Evaluation

We evaluate our proposed PointFlowMatch on RLBench [14], a popular robot learning benchmark. Each environment consists of the 7-DoF Franka Emika robot arm to execute actions and a set of five cameras around the workspace. We use RLBench's automatic expert demonstration execution to collect 100 trajectories for each task, which consists of all robot states and camera observations. For evaluation, we rollout the policy on each task for 100 episodes and report the average task success rate across all episodes. Following RLBench's test protocol, for each episode the task setup is randomized. We further perform three evaluation runs with three random seeds and average over all runs. As a result, all models are evaluated on the same set of 300 random episodes. Following ChainedDiffuser [10], we consider a set of eight tasks that require continuous interaction with the environment,

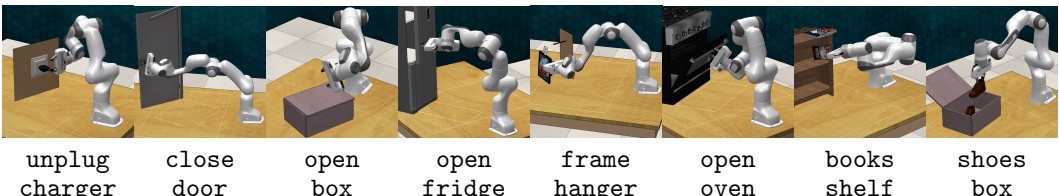

| unplug charger | close door | open box | open fridge | frame hanger | open oven | books shelf | shoes box |

Figure 2: Example images of the eight RLBench tasks.

Table 1: Performance comparison of PointFlowMatch with different baseline methods on the RLBench set of tasks. We report the success rate (SR) (↑) as well as the delta to our method. On average, our method performs considerably better. For OL-ChainedDiffuser we report the results from the original publication, which do not include the standard deviation.

| Method | unplug charger | close door | open box | open fridge | frame hanger | open oven | books shelf | shoes box | Mean SR | Delta SR |
|---|---|---|---|---|---|---|---|---|---|---|
| Dif. Policy [6] | $38.0^{\pm 3.6}$ | $19.3^{\pm 2.5}$ | $75.7^{\pm 4.2}$ | $0.0^{\pm 0.0}$ | $16.0^{\pm 2.6}$ | $0.0^{\pm 0.0}$ | $0.3^{\pm 0.6}$ | $0.0^{\pm 0.0}$ | $18.7^{\pm 2.3}$ | 49.1 |
| AdaFlow [15] | $46.3^{\pm 1.5}$ | $13.3^{\pm 3.1}$ | $77.3^{\pm 3.8}$ | $2.0^{\pm 0.0}$ | $12.7^{\pm 3.8}$ | $0.0^{\pm 0.0}$ | $0.3^{\pm 0.6}$ | $0.0^{\pm 0.0}$ | $19.0^{\pm 2.3}$ | 48.8 |
| 3D-DP [9] | $33.3^{\pm 4.7}$ | $\mathbf{76.0}^{\pm 1.7}$ | $98.3^{\pm 1.5}$ | $4.3^{\pm 2.1}$ | $12.3^{\pm 2.5}$ | $0.3^{\pm 0.6}$ | $3.7^{\pm 0.6}$ | $0.0^{\pm 0.0}$ | $28.5^{\pm 2.2}$ | 39.3 |
| OL-ChDif [10] | $65.0^{\pm \text{N/A}}$ | $21.0^{\pm \text{N/A}}$ | $46.0^{\pm \text{N/A}}$ | $\mathbf{37.0}^{\pm \text{N/A}}$ | $\mathbf{43.0}^{\pm \text{N/A}}$ | $16.0^{\pm \text{N/A}}$ | $40.0^{\pm \text{N/A}}$ | $9.0^{\pm \text{N/A}}$ | $34.6^{\pm \text{N/A}}$ | 33.2 |
| PointFlowMatch | $\mathbf{83.6}^{\pm 3.3}$ | $68.3^{\pm 6.6}$ | $\mathbf{99.4}^{\pm 0.7}$ | $31.9^{\pm 2.9}$ | $38.6^{\pm 2.7}$ | $\mathbf{75.9}^{\pm 4.0}$ | $\mathbf{68.8}^{\pm 5.8}$ | $\mathbf{76.0}^{\pm 3.5}$ | $\mathbf{67.8}^{\pm 4.1}$ | - |

where point-to-point motion planning approaches typically struggle: unplug_charger, close_door, open_box, open_fridge, take_frame_off_hanger, open_oven, put_books_on_bookshelf, and take_shoes_out_of_box (some names abbreviated in figures and tables).

## 4.1 Benchmarking Results

In this section, we compare the performance of PointFlowMatch with recent state-of-the-art methods for imitation learning based on generative models. For a fair comparison, we keep the number of denosining and/or integration steps $k$ constant to 50 for all methods.

For image-based methods, we compare against the original *Diffusion Policy* [6] as well as *AdaFlow* [15] which also uses the CFM learning objective. For the scope of this analysis, we do not implement the adaptive step feature in AdaFlow, since we mainly focus on the quality of prediction and not on inference speed. For both methods, we follow the original *Diffusion Policy* [6] approach and fuse images from all five viewpoints after the backbone feature extraction. As point cloud-based baselines, we consider *Chained Diffuser* [10] as well as *3D-DP* [9]. In this work, we specifically investigate the performance of non-hierarchical, single-task policies. For this reason, we consider the *Open loop trajectory diffusion* baseline presented in ChainedDiffuser [10], which is the underlying diffusion model without the higher level waypoint policy forming the hierarchy.

We report the results in Tab. 1. This evaluation highlights the strong performance of PointFlowMatch, which achieves a success rate 34 percentage points higher than the second-best model. This result demonstrates that the combination of our choices of observation type, encoder architecture, and training objective leads to a highly effective imitation learning algorithm. In line with the findings of our ablation study in Sec. 4.2, we also highlight how the point cloud based policies outperform the image-based policies.

## 4.2 Ablation Study

In this section, we aim to compare the key design choices of our method and answer the following research questions:

- Are point clouds a more effective observation representation compared to raw RGB images?
- Is Conditional Flow Matching a more effective training objective compared to the more established diffusion models?
- Is the $SO(3)$ formulation of CFM more effective at learning to match the data distribution on the 3D rotation manifold, compared to CFM in Euclidean space followed by a projection on $SO(3)$?

Table 2: Ablation of observation type (images vs point clouds), vector field formulation ($\mathbb{R}^6$ vs $SO(3)$), and training objective (DDIM vs CFM) for our method, evaluated on RLBench tasks showing the mean success rate (SR) ($\uparrow$) as well as the delta to the proposed model. All models are evaluated with $k = 50$ inference steps. Given the similar performance of the three point cloud-based baselines, we report the average across 3 training seeds, each tested on the same 3 evaluation seeds, for a total of 9 evaluation runs.

| Obs. type | Rot. formul. | Training objective | unplug charger | close door | open box | open fridge | frame hanger | open oven | books shelf | shoes box | Mean SR | Delta SR |
|---|---|---|---|---|---|---|---|---|---|---|---|---|
| Img. | $\mathbb{R}^6$ | CFM | $51.0^{\pm2.0}$ | $54.3^{\pm3.2}$ | $97.3^{\pm1.2}$ | $10.0^{\pm3.6}$ | $21.3^{\pm3.2}$ | $41.0^{\pm2.0}$ | $19.7^{\pm4.7}$ | $26.3^{\pm4.9}$ | $40.1^{\pm3.3}$ | $-27.7$ |
| Pcd. | $\mathbb{R}^6$ | DDIM | $\mathbf{84.8}^{\pm2.1}$ | $\mathbf{74.9}^{\pm5.3}$ | $99.3^{\pm0.9}$ | $30.8^{\pm3.7}$ | $\mathbf{39.1}^{\pm3.8}$ | $75.2^{\pm3.7}$ | $67.8^{\pm5.6}$ | $71.9^{\pm6.3}$ | $\mathbf{68.0}^{\pm4.3}$ | $0.2$ |
| Pcd. | $SO(3)$ | CFM | $82.3^{\pm3.1}$ | $68.8^{\pm4.8}$ | $\mathbf{99.4}^{\pm0.6}$ | $\mathbf{34.0}^{\pm7.6}$ | $38.4^{\pm4.0}$ | $74.9^{\pm2.0}$ | $68.0^{\pm6.4}$ | $73.2^{\pm1.2}$ | $67.4^{\pm4.4}$ | $-0.4$ |
| Pcd. | $\mathbb{R}^6$ | CFM | $83.6^{\pm3.3}$ | $68.3^{\pm6.6}$ | $\mathbf{99.4}^{\pm0.7}$ | $31.9^{\pm2.9}$ | $38.6^{\pm2.7}$ | $\mathbf{75.9}^{\pm4.0}$ | $\mathbf{68.8}^{\pm5.8}$ | $\mathbf{76.0}^{\pm3.5}$ | $67.8^{\pm4.1}$ | — |

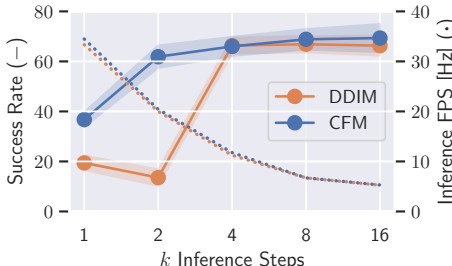

| $k$-steps | | 1 | 2 | 4 | 8 | 16 | 50 |
|---|---|---|---|---|---|---|---|
| Inference Time [ms] | DDIM | 30 | 50 | 89 | 149 | 190 | 362 |
| | CFM | 29 | 49 | 85 | 149 | 189 | 354 |
| Succes Rate | DDIM | 19.4 | 13.5 | 66.3 | 66.9 | 66.3 | 68.0 |
| | CFM | 36.8 | 61.9 | 66.0 | 68.8 | 69.3 | 67.8 |

Figure 3: Comparison of CFM and DDIM for varying values of the number of inference steps $k$. We compare the inference time ($\downarrow$) measured in [ms] as well as the inference FPS ($\uparrow$) in [Hz] against overall success rate ($\uparrow$) for both formulations.

- How does the number of inference steps $k$ affect performance?

In order to answer these questions, we carry out an ablation study where we vary each of the aforementioned design choices and investigate their impact on overall performance. In particular, we investigate 1) using images as visual observation, adopting the same ResNet backbone as in the original DP [6], 2) using the Denoising Diffusion Implicit Model (DDIM) [34] instead of the CFM training objective, 3) adopting the $SO(3)$ formulation of CFM for the end-effector orientations to learn probability paths directly on the 3D rotations manifold, 4) testing both the DDIM and CFM-based models with varying inference steps $k$, ranging from 1 to 16. We report the results in Tab. 2 and Fig. 3. From the numerical evaluation, we observe that the largest impact on performance is determined by the choice of observation type. The policy using raw RGB inputs is 27.7 percentage points worse than the proposed model, highlighting the strengths of point cloud observations.

First, we consider the two different formulations of the CFM framework that can be used to learn about the distribution of 3D rotations: euclidean and $SO(3)$. The outcome for this ablation is very similar, with the score averaged across all tasks showing that the $SO(3)$ formulation achieves a lower success rate, by 0.4 percentage points. This result is surprising and at first counterintuitive, as we expected that learning probability paths directly on the target manifold would achieve higher performance. Given the intriguing result of this ablation, we add further analysis of this comparison in Appendix A.1.

Regarding the choice of training objective, the results in Tab. 2 show that for $k = 50$ inference steps, the CFM objective does not improve over the popular DDIM framework, which achieves a similar success rate, within the standard deviation. Nevertheless, when considering the two objectives across a range of value of $k$ inference steps in Fig. 3, we find that CFM demonstrates stronger performance, especially in the low $k$ regime. This is in line with the results obtained for the image synthesis domain, where Stable Diffusion 3 [24] shows that CFM achieves similar scores to the diffusion objectives, but performs better with fewer inference steps, required for faster inference. This characteristic, paired with the higher flexibility, generality, and ease of implementation, in our opinion, makes the CFM formulation advantageous compared to established diffusion objectives.

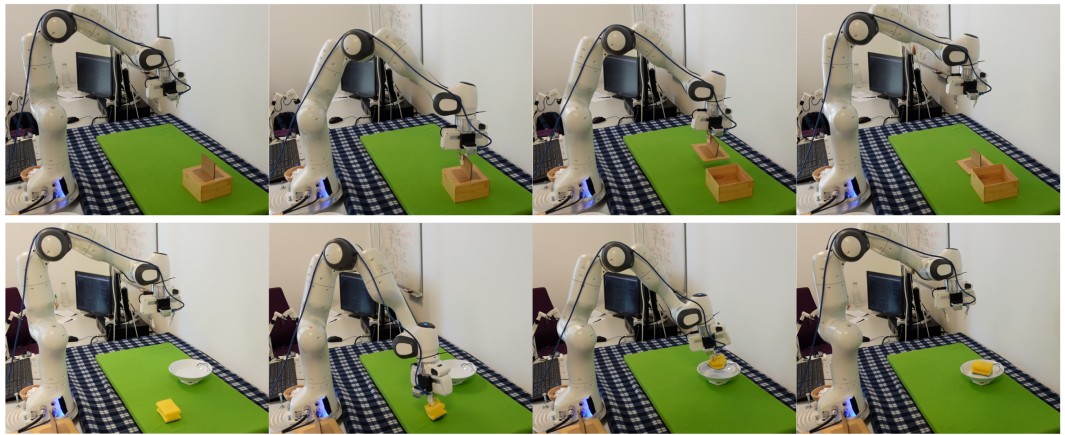

Figure 4: We demonstrate PointFlowMatch on a real robotic setup. We evaluate on two tasks: `open box` and `sponge on plate`.

### 4.3 Real Robot Experiments

To demonstrate the application of PointFlowMatch on a real robotic platform, we perform a set of experiments with a Franka Emika Panda manipulator. We evaluate on two table-top tasks: `open box` and `sponge on plate`, shown in Fig. 4. We use two Realsense RGB-D cameras, the first mounted on the robot end-effector and the second mounted externally. Similarly to the simulation setup, we merge the point clouds obtained from both cameras and process it via voxel-subsampling. To prevent overfitting and improve the model's robustness to the noisy point clouds, we apply random transformation and random noise-jitter augmentations during training. PointFlowMatch achieved a success rate of 72% for `open box` and 48% for `sponge on plate`. The main failure mode we observed in this experiment consisted of the robot reaching correctly for the object to grasp, but missing it by a small margin. A selection of policy rollouts for both tasks is shown in the accompanying video.

## 5 Conclusion

We present PointFlowMatch, a novel method for imitation learning from a fixed set of demonstration examples. Our method combines the recently developed conditional flow matching framework with a point cloud observation encoding. In developing our method, we performed an ablation study of the most important components. Additionally, we study a formulation of CFM on the SO(3) manifold. Our method achieves state-of-the-art results for single-task non-hierarchical policy learning, with an average success rate of 68.6 %, double that of the closest baseline method.

**Limitations:** There are a few limitations to our proposed method. Compared to approaches such as BC that directly predict an action, diffusion and CFM models both have longer inference times, given their iterative nature. In addition to this, as usual in the fixed-data imitation learning setting, CFM cannot extrapolate out of distribution and thus, only learns motion correction behavior when included in the demonstration set. Our point cloud observation space merges multiple views, this requires depth cameras with known intrinsic and extrinsic calibration. We also trained single-task policies and did not investigate generalization capabilities. Finally, implementation on a real robot requires manually collecting expert demonstrations.

**Acknowledgments**

This work was funded by the Carl Zeiss Foundation with the ReScaLe project and the German Research Foundation (DFG): 417962828.

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

# A    Additional Experiments

## A.1    Simplified $SO(2)$-Experiment: No Free Lunch

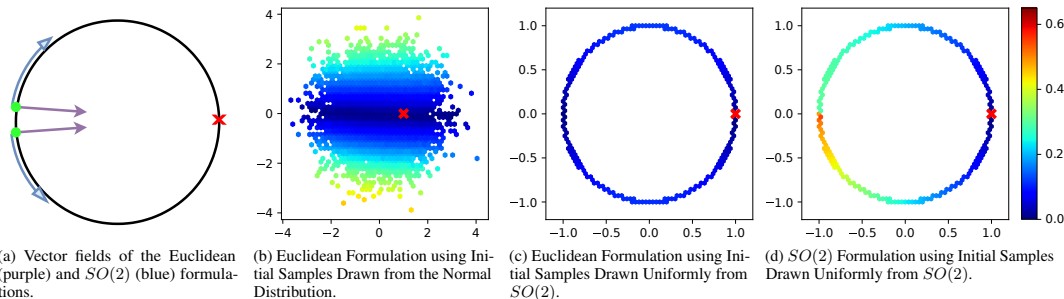

(a) Vector fields of the Euclidean (purple) and $SO(2)$ (blue) formulations.

(b) Euclidean Formulation using Initial Samples Drawn from the Normal Distribution.

(c) Euclidean Formulation using Initial Samples Drawn Uniformly from $SO(2)$.

(d) $SO(2)$ Formulation using Initial Samples Drawn Uniformly from $SO(2)$.

Figure 5: Simplified Example. The left figure shows the edge case when random samples are close to the opposite pole of the target sample. Here the $SO(3)$ *formulation* presents a discontinuity which makes learning more difficult. In the three right figures, we visualize the mean error during inference across different sampling locations for our different formulations. We mark the target with a red cross. One observes that for the *Euclidean formulation* the error is lower for initial sample points along the axis orthogonal to the target. This is expected as values sampled along the line are naturally mapped to the target when normalized. On the other side in the last figure, one observes higher errors close to the pole. Additionally, a training data bias is visible as the error is higher on one side of the discontinuity.

To further investigate the counterintuitive result discussed in Sec. 4.2, we set up a toy task on the unit circle. Here, the goal is to infer a rotation of 0 degrees, or in other words the coordinate at $(1,0)$ on the unit circle. For this experiment, the prediction is not conditioned, and all samples are regressed against the unique target point $(1,0)$. As done in Sec. 4.2, we evaluate two formulations, one where the target vector field is defined in the Euclidean space $\mathbb{R}^2$, and one where it follows the geodesics on the Riemannian manifold $SO(2)$ [30]. In the following, we refer to them as *Euclidean formulation* and $SO(2)$ *formulation*. We provide an overview of the formulations (robotic and toy experiments) in Tab. 3.

As in the case of $SO(3)$ (see Sec. 3.1), the input to the vector field prediction network is the truncated version of the full rotation state matrix, i.e. a 2D vector. We train both formulations on batches of size 1000 for 2000 epochs using the Adam optimizer with cosine annealed learning rate of $1e^{-3}$. For evaluations, we evaluate 10000 noise samples and perform 50 integration steps. Given the final state, we calculate the absolute angle error to the target $(1,0)$.

The *Euclidean formulation* achieves an average angle error of $0.093°$ and the $SO(2)$ *formulation* $0.216°$. We visualize the mean error across initial sample points in Fig. 5. We conclude that both formulations can successfully solve the task, but we also observe higher final angle error for the $SO(2)$ *formulation*, similar to the ablation study from Sec. 4.2.

As shown in Fig. 5a, when random samples are close to the opposite pole of the target sample, the $SO(2)$ case presents strong discontinuities, with vector fields of nearby points pointing in opposite directions. For successful gradient-based learning, however, we need the input-output mapping to exhibit some notion of continuity. The exponential coordinates used to represent vectors on the tangent space do not fulfill the pre-images connectivity constraint, which means that a smooth function to interpolate between input samples is not guaranteed to exist [29, 35]. A discontinuity in the target mapping results in difficulties in fitting the model via gradient-based learning. Additionally, if the training data for the network is not equally distributed on both sides of the pole, we introduce a data bias towards either side.

For the *Euclidean formulation* on the other hand, the need to project the final output to the manifold of valid rotations is also a source of error, since the projection is only applied at inference time and it is not part of the training process. This can be seen in Fig. 5b, where random samples further away from the x-axis present higher absolute angle errors. As a result, we conclude that both formulations exhibit different characteristics, each with their drawbacks, and the choice of one option over the other needs to be evaluated carefully.

Table 3: Comparison of different state dimensions and their respective velocities, target calculation and progression formulations. For definitions of the Log($\bullet$)- and Exp($\bullet$)-maps we refer to Sola *et al.* [30]. *[†]As for calculating the final angle only the first column is needed, we omit calculating the second orthonormal vector. Thus, instead of performing the full Grahm-Schmidt process, we only perform a normalization of the inferred state variable that maps the vector to the unit circle.*

| Formulation | State $z$ | Velocity $v$ | Target Velocity $v_{\text{gt}}$ | State Progression | State Conversion |
|---|---|---|---|---|---|
| 3D-Euclidean | $\mathbb{R}^6$ | $\mathbb{R}^6$ | $z_1 - z_0$ | $z + \Delta t \, v$ | Grahm-Schmidt |
| $SO(3)$ | $\mathbb{R}^{3\times3}$ | $\mathbb{R}^3$ | $\text{Log}_{SO(3)}\left(z_0^{-1}z_1\right)$ | $z \, \text{Exp}_{SO(3)}\left(\Delta t \, v\right)$ | N/A |
| 2D-Euclidean | $\mathbb{R}^2$ | $\mathbb{R}^2$ | $z_1 - z_0$ | $z + \Delta t \, v$ | Grahm-Schmidt[†] |
| $SO(2)$ | $\mathbb{R}^{2\times2}$ | $\mathbb{R}^1$ | $\text{Log}_{SO(2)}\left(z_0^{-1}z_1\right)$ | $z \, \text{Exp}_{SO(2)}\left(\Delta t \, v\right)$ | N/A |

