# OpenReview forum: "Learning Robotic Manipulation Policies from Point Clouds with Conditional Flow Matching"
_robot-learning.org/CoRL/2024/Conference — CoRL 2024_

### Official Review · Reviewer_mSNh · 2024-07-17
**Contribution is unclear and the results are not strong enough.**

**Originality:** 3
**Technical Quality:** 3
**Clarity Of Presentation:** 3
**Potential Impact:** 3
**Recommendation:** 3
**Confidence:** 4

**Review:**

Pros
- The authors describe very clearly the input and output of the models, the hyper parameters, and number of training iterations. These details are helpful for reproducibility and to compare with other models.
- The visualization in Fig. 3 (c) (d) is interesting. The discontinuity can be clearly seen from the figure.
- The ablation is helpful.

Cons
- The contribution is not clear enough. The use of point cloud has been studied in many other papers. The continuity on rotational representation is presented in [26]. The major contribution to me is using CFM instead of diffusion models. However, the theoretical advantage of CFM is not clear to me. Though there is a potential flexibility in encoding the distribution, it is not illustrated clearly through the writing or experiments. According to Table 2, using CFM only has a marginally higher SR than DDIM, which renders the main contribution less appealing.
- I think the continuity of SO3/SO2 representation has been well studied in [26]. Is the part comparing 6D and tangential representation not covered there? Please correct me if I’m wrong.

Other
- P3 L110 - ’T_{obs} is a tuning hyperparameter’. Looks like there were some equations here and results in a typo?
- About the paragraph above Sec. 5. I’m not quite following here.  Should there be another plot for the SO3 representation to draw the conclusion?


Overall I think either 1) more analysis be provided on what theoretical advantage CFM has in IL, or 2) more convincing empirical result be presented, would make this a stronger paper.

**Quality Of The Limitations Section:**

3

**Questions For Rebuttal:**

- Some result numbers are really close. I’m curious about the significance of the result. It’ll be great if there are some measure of variance there.
- Are all baseline models trained using the same data generated by the authors?

**Robotics Focus:**

3

**Summary Of Paper:**

This paper applies conditional flow matching (CFM) to imitation learning. CFM is a more general form of generative models. The method is evaluated against baselines using diffusion models. The authors also investigates the use of 6D representation and SO3 representation in the rotation matrix. The third emphasis of the work is using point cloud as input.

**Summary Of Recommendation:**

Trying different generative models is nice. But there is only limited discussion on why would using CFM have advantages in the context of robotics.  Results are not strong enough. After the rebuttal: Though the results are not very strong, there is some potential benefits of the framework. The conclusion on rotation formulation could be helpful to other authors. I'm inclined to accept the paper.

---

### Official Review · Reviewer_s1fJ · 2024-07-21
**Conditional flow matching from pointclouds**

**Originality:** 2
**Technical Quality:** 2
**Clarity Of Presentation:** 3
**Potential Impact:** 3
**Recommendation:** 3
**Confidence:** 2

**Review:**

- CNF typically refers to continuous normalizing flows, not conditional normalizing flows.

- When listing disadvantages of diffusion models you state: "the space of sampling probability paths is limited (only Gaussian)". This is not necessarily true. Many existing diffusion works use non-gaussian noise [1]

- "which restricts 39 learning on simple manifolds and results in longer training times". This claim is also untrue. Diffusion has been applied to manifolds outside of standard euclidian [2].

- No supplementary materials? The paper says code, models, and videos are available but the link is dead.

- The results look really good in comparison with baselines, but no standard deviation were reported on the results and no statistical tests were performed. Given such a large improvement over baselines, I'm guessing the results are significant, but it would be good to know for certain.

- Section 4 says "Each result reported is the average across three random seed". This statement needs clarification. Is this three separate training seeds or three environment seeds? The policy is run for 100 episodes. Are the locations/positions of the objects randomized for each episode?

- Figure 3 might not be necessary. I think it is widely understood why training with euclidian distances on a non-euclidian space is a bad idea, and it would be easier to just point to one of the many recent works on this subject. It also seems not too relevant for this problem setting according to the results.

- No experiments performed on real robot setups. This is especially important here because depth images are notoriously noisy and are likely to significantly impact performance. One of the key questions the authors try to answer is "Are point clouds a more effective observation representation compared to raw RGB images?". They certainly contain more information than raw RGB images, but I think it is premature to make the claim that pointclouds are strictly better without testing on real-world pointclouds.

- I'm a bit confused by the results and discussion in this paper. The authors appear to be concluding that the choice of flow/diffusion and se3/euclidian is not very important and the choice of pointcloud/pointnet vs rgb/cnn has a much larger impact. Then why does 3D-DP perform so poorly on the RL bench problems if it is essentially an pointcloud-based diffusion model in euclidian space?

- The conclusion states that CFM has longer inference times? How much longer? Inference times should be reported somewhere.

[1] Bansal, A., Borgnia, E., Chu, H. M., Li, J. S., Kazemi, H., Huang, F., Goldblum, M., Geiping, J., & Goldstein, T. (2022). Cold Diffusion: Inverting Arbitrary Image Transforms Without Noise

[2] Elhag, A. A., Wang, Y., Susskind, J. M., & Bautista, M. A. (2023). Manifold Diffusion Fields.

**Quality Of The Limitations Section:**

3

**Questions For Rebuttal:**

See above

**Robotics Focus:**

3

**Summary Of Paper:**

This paper introduces PointFlowMatch, a method combining Conditional Flow Matching with point cloud observations for robotic manipulation learning. Evaluated on RLBench tasks, it outperforms existing methods. The authors analyze various design choices, concluding that point cloud inputs significantly contribute to the method's effectiveness while other strategies such as manifold learning and normalizing flows do not.

**Summary Of Recommendation:**

The paper presents a novel combination of existing methods in generative modeling and pointcloud neural networks for behavior cloning. While the architecture is novel, the experiments are lacking real-world evaluations. Additionally, no supplementary materials were provided.

---

### Official Review · Reviewer_E1uS · 2024-07-21
**Excellent investigation of conditional flow matching as an alternative to diffusion policy.**

**Originality:** 3
**Technical Quality:** 4
**Clarity Of Presentation:** 4
**Potential Impact:** 3
**Recommendation:** 3
**Confidence:** 4

**Review:**

Strengths:
* The paper topic is pertinent to the research community as diffusion policy and its variants have become a critical research direction in robot learning.
* The literature review is thorough and does a good job contextualizing the paper in the literature.
* The comparisons are extensive, consisting of many SOTA diffusion policy-style methods and different input-feature modalities.
* The discussion is insightful. It is good to see a fair investigation that finds certain choices important (poinct cloud features) and others providing only modest gains (CFM vs diffusion).

Weaknesses:
* The evaluation is only in simulation on RLBench; there are no hardware experiments.
* The literature review is missing a few concurrent works:

[A] Braun, M., Jaquier, N., Rozo, L., & Asfour, T. (2024). Riemannian Flow Matching Policy for Robot Motion Learning. arXiv preprint arXiv:2403.10672. (Came out before submission.)

[B] Rouxel, Q., Ferrari, A., Ivaldi, S., & Mouret, J. B. (2024). Flow Matching Imitation Learning for Multi-Support Manipulation. arXiv preprint arXiv:2407.12381. (Came out after submission.)

* In the limitations, it is not true that imitation learning can't learn 'motion corrective behaviors'. It is possible to learn them if they are included in the demonstrations in small quantities.

* The following is a non-exhaustive list of typos and grammatic errors.
1. Line 129: 'supervising' -> 'supervision'.
2. Line 284: there should be no comma after diffusion.
3. Make sure to capitalize acronyms (e.g., 3D) in the paper titles of the references appropriately.

**Quality Of The Limitations Section:**

2

**Questions For Rebuttal:**

In addition to the weaknesses listed above, I have the following questions for the authors.
1. In Sec. 3.2, why use a Gaussian distribution if one of the advantages is the generality of the possible distributions? It would be interesting to explore other possibilities.
2. For my understanding, why is the linear step in Algorithm 1 taken before the network during training and after during inference?

**Robotics Focus:**

3

**Summary Of Paper:**

The paper presents an investigation of conditional flow matching (CFM) versus diffusion policy. The authors also ablate image versus point cloud features and different ways to learn rotations. The experiments consider several SOTA imitation learning architectures on RLBench, showing better performance for CFM and 3D point cloud features as compared to diffusion policy on image-based features.

**Summary Of Recommendation:**

Overall, I think this is a solid paper accomplishing what it sets out to do in terms of architecture investigations. The use of CFM as compared to diffusion policy is promising as it is more general and does not require both a backward and forward pass. The only concern with this paper that I have is the lack of hardware experiments.

---

### Author Rebuttal · Authors · 2024-08-14

Thank you for your thoughtful feedback and valuable comments. We have carefully revised the paper to address all the concerns and suggestions of the reviewers. A revised pdf is now uploaded with new additions marked in blue, as well as a video demonstrating our latest robot experiments. As per the FAQ of the rebuttal (https://docs.google.com/document/d/1I2gXN_iVWspGKAgN6GmTl3SCPNgScej4nPF0QCefEBE/edit), we allow ourselves to submit this revised version over the page limit.
We really hope that the reviewers see the value in our work and consider raising their scores. Thank you for your invaluable time and consideration!

---

### Decision · Program_Chairs · 2024-09-04

**Decision:**

Accept

**Comment:**

This paper introduces PointFlowMatch, a method that combines Conditional Flow Matching (CFM) with point cloud observations for imitation learning in robotic manipulation. The authors demonstrated improved performance compared to baselines on RLBench tasks.

Before the rebuttal, the reviewers acknowledge the importance and relevance of the proposed work but raised several concerns. The primary issue is that the results are not strong enough with the performance only demonstrated through simulation on RLBench. There are no hardware experiments to validate the method in more complicated real-world scenarios. Additional concerns include a lack of rigorous analysis on the advantages of CFM in the context of robotics, missing supplemental materials that were promised, and the absence of discussions on more related works, among others.

During the rebuttal, the authors did an excellent job addressing the reviewers’ concerns, particularly by including hardware results. As a result, all reviewers now rate the paper as a Weak Accept. However, the reviewers mentioned during the discussion that concrete comparisons between diffusion and flow methods will be necessary to further enhance the impact and relevance of the work to the robot learning community.